==============================

# Generating Multi-Sentence Abstractive Summaries of Interleaved Texts

## Abstract

In multi-participant postings, as in online chat conversations, several conversations or topic threads may take place concurrently. This leads to difficulties for readers reviewing the postings in not only following discussions but also in quickly identifying their essence. A two-step process, disentanglement of interleaved posts followed by summarization of each thread, addresses the issue, but disentanglement errors are propagated to the summarization step, degrading the overall performance. To address this, we propose an end-to-end trainable encoder-decoder network for summarizing interleaved posts. The interleaved posts are encoded hierarchically, i.e., word-to-word (words in a post) followed by post-to-post (posts in a channel). The decoder also generates summaries hierarchically, thread-to-thread (generate thread representations) followed by word-to-word (i.e., generate summary words). Additionally, we propose a hierarchical attention mechanism for interleaved text. Overall, our end-to-end trainable hierarchical framework enhances performance over a sequence to sequence framework by 8-10% on multiple synthetic interleaved texts datasets.

## Introduction

Interleaved texts are becoming more common with new ways of working and new forms of communication, e.g., multi-author entries for activity reports, meeting minutes and social media conversations, such as Slack. Quickly getting a sense of or following the content of different threads in interleaved texts, where posts belonging to different threads occur in one sequence, is often difficult. An example of two threads with multiple posts interleaved to form a sequence is:

> Post1-Thread1 → Post1-Thread2 → Post2-Thread1 → Post3-Thread1 → Post2-Thread2 → Post3-Thread2

This intermingling leads to difficulties in not only following discussions but also in retrieving their essence. In conversation disentanglement, interleaved posts are grouped by the thread; e.g., the previous example could be rearranged as:

> Post1-Thread1 → Post2-Thread1 → Post3-Thread1
> Post1-Thread2 → Post2-Thread2 → Post3-Thread2

In analyzing interleaved texts, Shang et al. (2018) went a step further and proposed summarization of the interleaved texts. They designed an unsupervised two-step system and evaluated the system on meeting texts. In the first step, a conversation disentanglement component disentangles the texts thread-wise. Then, in the second step, a multi-sentence compression component compresses the thread-wise posts to single sentence summaries. However, this system has a major disadvantage, in that the disentanglement obtained through either supervised Jiang et al. (2018) or unsupervised Wang & Oard (2009) methods propagate its errors to the downstream summarization task, and thus, degrades the overall performance.

We aim to tackle this issue of error propagation through an end-to-end trainable encoder-decoder system that takes a variable length input, e.g., interleaved texts, processes it and generates a variable length output, e.g., a multi-sentence summary; see Figure 1 for an illustration. An end-to-end system eliminates the disentanglement component, and thus, the error propagation.

Figure 1: 7 interleaved posts are implicitly disentangled into 3 threads, and single sentence summaries are generated for each thread. Posts are outlined with colors corresponding the color of each summary.

The encoder first performs word-to-word encoding to embed each post, followed by post-to-post encoding to embed the overall content of the posts and represent the discourse structure of the interleaved texts. The decoder has a thread-to-thread decoder to generate a representation for each thread, and the thread representation is given to a word-to-word decoder to generate a summary sentence. We also propose to use hierarchical attention similar to Nallapati et al. (2016), but instead of computing post-level attention at every word, attentions are only computed when decoding new sentences. Further, the attention networks are trained end-to-end.

Despite the availability of a multitude of real-world interleaved texts, a major challenge to train encoder-decoder models is the lack of labels (summaries). As labeling is difficult and expensive Barker et al. (2016); Aker et al. (2016); Verberne et al. (2018), we synthesize two separate corpora. Each is derived by mixing texts and associated summaries from a corpus of documents, where the mixed text has a structure reflective of a multi-party conversation with inter-leaved threads and the summary highlights the information in the threads. We find abstracts and titles of randomized controlled trial (RCT) articles, a PubMed corpus, and also for Stack Exchange posts and titles.

To summarize, our contributions are threefold:

- We propose to combine a hierarchical encoder and decoder to obtain multi-sentence summaries of interleaved texts.
- We propose a novel hierarchical attention mechanism that is integrated with the hierarchical encoder-decoder architecture and which equips the decoder to disentangle the interleaving further.
- We use two synthetic datasets to verify the ideas and show our end-to-end trainable architecture addresses not only the issue of error-propagation observed in competitive methods but also improves the performance.

## RELATED WORK

Quite often multi-party conversations, e.g. news comments, social media conversation and activity report, have tens of posts discussing several different matters pertaining to a subject. Ma et al. (2012); Aker et al. (2016); Shang et al. (2018) designed methodologies to summarize posts in order to provide an overview on the discussed matters. They broadly follow the same approach: cluster the posts and then extract a summary from each cluster.

There are two kinds of summarization: abstractive and extractive. In abstractive summarization, the model utilizes a corpus level vocabulary and generates novel sentences as the summary, while extractive models extract or rearrange the source words as the summary. Abstractive models based on neural sequence-to-sequence (seq2seq) (Rush et al. (2015)) proved to generate summaries with higher ROUGE scores than the feature-based abstractive models. Integration of attention into seq2seq

(Bahdanau et al. (2014)) led to further advancement of abstractive summarization (Nallapati et al. (2016); Chopra et al. (2016)).

There are many possible patterns of organization of the information in texts, e.g., chronological pattern. News articles have an inverted pyramid pattern, i.e., core information in the lead sentences and the extra information in later sentences. A seq2seq model is appropriate for summarization of a news article as it encodes and decodes sequentially. However, in interleaved texts, related information maybe separated; thus a seq2seq model may be competent but not sufficient.

Li et al. (2015) proposed an encoder-decoder (auto-encoder) model that utilizes a hierarchy of networks: word-to-word followed by sentence-to-sentence. Their model is better at capturing the underlying structure than a vanilla sequential encoder-decoder model (seq2seq). Krause et al. (2017); Jing et al. (2018) showed multi-sentence captioning of an image through hierarchical Recurrent Neural Network (RNN), topic-to-topic followed by word-to-word, is better than seq2seq.

These works suggest a hierarchical encoder, with word-to-word encoding followed by post-to-post, will better recognize the dispersed information in interleaved texts. Similarly, a hierarchical decoder, thread-to-thread followed by word-to-word, will intrinsically disentangle the posts, and therefore, generate more appropriate summaries.

Nallapati et al. (2016) devised a hierarchical attention mechanism for a seq2seq model, where two levels of attention distributions over the source, i.e., sentence and word, are computed at every step of the word decoding. Based on the sentence attentions, the word attentions are rescaled. Hsu et al. (2018) slightly simplified this mechanism and computed the sentence attention only at the first step. Our hierarchical attention is more intuitive and computes new sentence attentions for every new summary sentence, and unlike Hsu et al. (2018), is trained end-to-end .

## MODEL

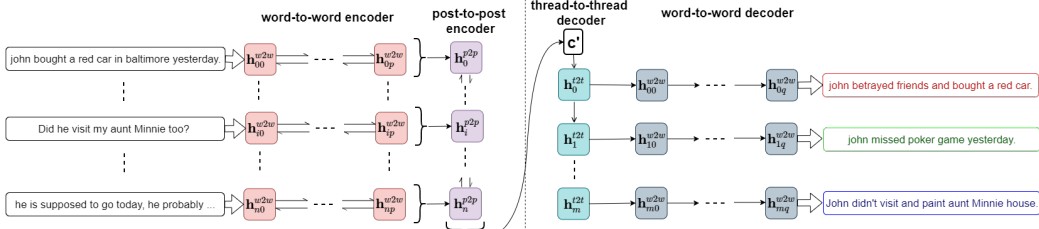

Figure 2: Hierarchical encoder-decoder architecture.

### PROBLEM STATEMENT

We aim to design a system that when given a sequence of posts, $C = \langle P_1, \ldots, P_{|C|} \rangle$, produces a sequence of summaries, $T = \langle S_1, \ldots, S_{|T|} \rangle$. For simplicity and clarity, unless otherwise noted, we will use lowercase italics for variables, uppercase italics for sequences, lowercase bold for vectors and uppercase bold for matrices.

Figure 2 illustrates the proposed hierarchical encoder-decoder framework. In the framework, first, a low-level, word-to-word encoder converts a sequence of words in a post, $P_j$, to a sequence of representations, $H_j = \langle \mathbf{h}_{j0}^{w2w}, \ldots, \mathbf{h}_{j|P_j|}^{w2w} \rangle$. Subsequently, a top-level, post-to-post encoder converts those representations, $\langle H_0, \ldots, H_{|C|} \rangle$, to a sequence of top-level post representations $\langle \mathbf{h}_1^{p2p}, \ldots, \mathbf{h}_{|C|}^{p2p} \rangle$. These encoded representations are then passed to the decoder, which utilizes a top-level, thread-to-thread, decoder to disentangle them into a sequence of thread representations $\langle \mathbf{h}_1^{t2t}, \ldots, \mathbf{h}_{|T|}^{t2t} \rangle$. Finally, a low-level, word-to-word, decoder takes a feed-forward mapped thread representation $\mathbf{h}_i^{t2t}$ and generates a sequence of summary words $\langle w_{i1}, \ldots, w_{i|S_i|} \rangle$.

The maximum number of posts in a sequence of interleaved texts is denoted by $n$ and threads by $m$. We limit the number of words in posts and summaries to fixed lengths by either truncating or

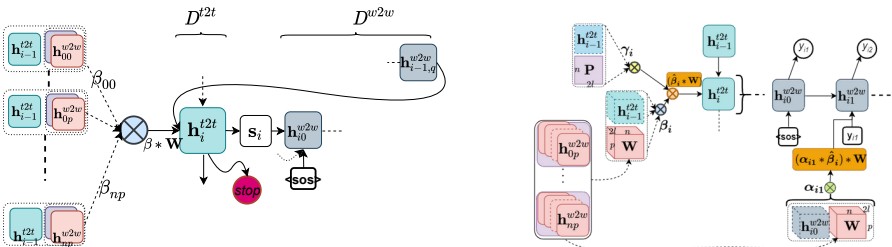

Figure 3: A step in the thread-to-thread decoder. Figure 4: Hierarchical attention mechanism for a hierarchical encoder.

padding, and denote them by $p$ and $q$ respectively. The hidden states of the encoder and decoder have dimensionality $l$.

### ENCODER

The hierarchical encoder used in Figure 2 is based on Nallapati et al. (2017), where word-to-word and post-to-post encoders are bi-directional LSTMs. The word-to-word Bi-LSTM encoder ($E_{w2w}$) runs over word embeddings of any post, $P_j$, and generates a set of hidden representations ($\langle \mathbf{h}_{j0}^{E_{w2w}}, \ldots, \mathbf{h}_{jp}^{E_{w2w}} \rangle$); see left middle section in Figure 2. The average pooled value of the word-to-word representations of post $P_j$ ($\frac{1}{p} \sum_{k=0}^{p} \mathbf{h}_{jk}^{E_{w2w}}$) are input to the post-to-post Bi-LSTM encoder ($E_{t2t}$), which then generates a set of representations ($\langle \mathbf{h}_{0}^{E_{p2p}}, \ldots, \mathbf{h}_{n}^{E_{p2p}} \rangle$) corresponding to those posts. Overall, output word-to-word representations, $\mathbf{W}$, has dimension $n \times p \times 2l$ and post-to-post representations, $\mathbf{P}$, has dimension $n \times 2l$ for a given channel, $C$.

### DECODER

Our hierarchical decoder (see Figure 2) is based on Li et al. (2015), where both thread-to-thread and word-to-word decoders are uni-directional LSTMs.

The initial state $\mathbf{h}_{0}^{D_{t2t}}$ of the thread-to-thread LSTM decoder ($f^{D_{t2t}}$) is set with a feedforward-mapped representation of an average pooled post representations ($\mathbf{c}' = \frac{1}{n} \sum_{k=0}^{n} \mathbf{h}_{k}^{p2p}$). At each step $i$ of the decoder, a sequence of attention weights, $\langle \hat{\beta}_{i,0}, \ldots, \hat{\beta}_{i,n \times p} \rangle$, corresponding to the set of encoded word representations, $\langle \mathbf{h}_{0}^{w2w}, \ldots, \mathbf{h}_{n \times p}^{w2w} \rangle$ are computed utilizing the previous state, $\mathbf{h}_{i-1}^{D_{t2t}}$. We will elaborate the attention computation in the next section.

A weighted representation of the words (crossed blue circle) is then computed: $\sum_{j=1}^{n} \hat{\beta}_{ij} \mathbf{W}_{ij}$, and used as one input to LSTM $f^{D_{t2t}}$. Additionally, we use the last hidden state $\mathbf{h}_{i-1}^{D_{w2w}}$ of the word-to-word decoder LSTM ($D_{w2w}$) of the previously generated summary sentence as the second input to compute the next state of thread-to-thread decoder, i.e., $\mathbf{h}_{i}^{D_{t2t}}$. The motivation is to provide information about the previous sentence.

The current state $\mathbf{h}_{i}^{D_{t2t}}$ is passed through a single layer feedforward network and a distribution over STOP=1 and CONTINUE=0 is computed:

$$p_i^{STOP} = \sigma(\mathbf{g}\left(\mathbf{h}_i^{D_{t2t}}\right)) \tag{1}$$

where g is a feedforward network. In Figure 3, the process is depicted by a red circle. The thread-to-thread decoder keeps decoding until $p_i^{STOP}$ is greater than 0.5.

Additionally, the current state $\mathbf{h}_{i}^{D_{t2t}}$ is passed through another single layer feedforward network $k$ followed by $\tanh$ activation to compute the thread representation $\mathbf{s}_i = \tanh(\mathbf{k}\left(\mathbf{h}_i^{D_{t2t}}\right))$.

Given a thread representation $\mathbf{s}_i$, the word-to-word decoder generates a summary for the thread. Our word-to-word decoder is based on Bahdanau et al. (2014). It is a unidirectional attentional LSTM ($f^{D_{w2w}}$); see the right-hand side of Figure 2. We refer to Bahdanau et al. (2014) for further details.

HIERARCHICAL ATTENTION

Our novel dynamic hierarchical attention is based on two popular concepts: attention scores should integrate in top-down manner (Nallapati et al. (2016)) and two distributions over same words must be modeled together (See et al. (2017)), not just in isolation.

Our hierarchical attention works at 3 levels, the post level (corresponding to posts), i.e., $\boldsymbol{\gamma}$, and higher level (corresponding to source tokens), i.e., $\boldsymbol{\beta}$, are computed while obtaining a thread representation, $\mathbf{s}$. The lower level attention (also corresponding to source tokens), i.e., $\boldsymbol{\alpha}$, is computed while generating a word, $y$, of a summary, $S$.; see yellow rectangles in Figure 4.

At step $i$ of thread decoding, we also compute elements of post-level attention, i.e., $\boldsymbol{\gamma}_{i\cdot}$ as below.

$$\gamma_{ij} = \sigma(\text{attn}^\beta(\mathbf{h}_{i-1}^{D_{t2t}}, \mathbf{P}_j) \quad j \in \{1, \ldots, n\} \tag{2}$$

At the same step, we also compute elements of high-level attention, i.e, $\boldsymbol{\beta}_{i\cdot}$ as below.

$$\beta_{ik} = \sigma(\text{attn}^\beta(\mathbf{h}_{i-1}^{D_{t2t}}, \mathbf{a}_k)) \quad k \in \{1, \ldots, n \times p\} \tag{3}$$

$$\mathbf{a}_k = \text{add}(\mathbf{W}_{jl}, \mathbf{P}_j) \text{ where } k := j * l \text{ and } j \in \{1, \ldots, n\}, \ l \in \{1, \ldots, p\} \tag{4}$$

where $\mathbf{a}_k$ is computed as in Eq. 4. $\mathbf{W}$ in Eq. 4 is a word-to-word encoder representations of dimension $n \times p \times 2l$, add aligns a post representation to its constituting word representations and does element-wise addition, and $attn^\beta$ is a feedforward network that aligns the current thread decoder state $\mathbf{h}_{i-1}^{D_{t2t}}$ with all $n * p$ number of $\mathbf{a}_k$ vectors.

Then, we use $\boldsymbol{\gamma}$ to rescale high-level attention, $\boldsymbol{\beta}$ as below.

$$\hat{\beta}_{ik} = \beta_{ik} * \gamma_{ij} \text{ where } k := j * l \text{ and } l \in \{1, \ldots, p\} \ j \in \{1, \ldots, n\} \tag{5}$$

At step $t$ of word-to-word decoding of summary thread $i$, we compute elements of low level attention, i.e., $\boldsymbol{\alpha}_{it\cdot}$ as below.

$$\alpha_{itk} = \frac{\exp(\mathbf{e}_{itk})}{\sum_{k=1}^{n \cdot p} \exp(\mathbf{e}_{itk})} \text{ where } \mathbf{e}_{itk} = \text{attn}^\alpha(\mathbf{h}_{t-1}^{D_{w2w}}, \mathbf{a}_k) \tag{6}$$

, and $\mathbf{a}_k$ is same as in Eq. 4 and $attn^\alpha$ is a feedforward network that aligns the current word decoder state $\mathbf{h}_{t-1}^{D_{w2w}}$ with all $n * p$ number of $\mathbf{a}_k$ vectors.

Finally, we use rescaled high-level word attentions, $\hat{\boldsymbol{\beta}}$, for rescaling low level attention, $\alpha$ as below:

$$\hat{\alpha}_{itk} = \frac{\hat{\beta}_k \times \exp(\mathbf{e}_{itk})}{\sum_{k=1}^{n \cdot p} \hat{\beta}_k \times \exp(\mathbf{e}_{itk})} \tag{7}$$

Table 1 compares our hierarchical attention with previous work on hierarchical attention in the text summarization domain.

Table 1: Forms of hierarchical attention. End2End=No training labels, $\times words$=for every word in the target, $\times summaries$=for every summary in the target, $\boldsymbol{\gamma}$=sentence(post)-level, $\boldsymbol{\beta}$=high-level

| Model | End2End | Decoding | $\boldsymbol{\gamma}$ computations | $\boldsymbol{\beta}$ |
|---|---|---|---|---|
| Nallapati et al. (2016), Cheng & Lapata (2016), Tan et al. (2017) | yes | seq | $\times words$ | no |
| Hsu et al. (2018) | no | seq | 1 | no |
| Ours | yes | hier | $\times summaries$ | yes |

TRAINING OBJECTIVE

We train our hierarchical encoder-decoder network similarly to an attentive seq2seq model Bahdanau et al. (2014), but with an additional weighted sum of sigmoid cross-entropy loss on stopping distribution; see Eq. 1. Given a summary, $Y_i = \langle w_{i0}, \ldots, w_{iq} \rangle$, our word-to-word decoder generates a target $\hat{Y} = \langle y_{i0}, \ldots, y_{iq} \rangle$, with words from a same vocabulary $U$. We train our model end-to-end by minimizing the objective given in Eq. 8.

$$\sum_{i=1}^{m} \sum_{j=1}^{q} \log p_\theta \left(y_{ij} | w_{i\cdot < j}, \mathbf{W}\right) + \lambda \sum_{i=1}^{m} y_i^{STOP} \log(p_i^{STOP}) \tag{8}$$

Table 2: The left rows contain interleaving of 3 articles with 2 to 5 sentences and the right rows contain their interleaved titles. Associated sentences and titles are depicted by similar symbols.

| |
|---|
| ✓ this study was conducted to evaluate the influence of e... 
 ✦ to assess the effect of a program of supervised fitness... 
 ✦ an 8-week randomized , controlled trial .... 
 ✓ nine endurance-trained athletes participated in a randomised... 
 ... 
 ✱ we examined the effects of intensity of training on ratings... 
 ✱ subjects were recruited as sedentary controls or were randomly... 
 ✱ the at lt group trained at velocity lt and the greater than... 
 ✓ data were obtained on 47 of 51 intervention patients and 45... |

| |
|---|
| ✓ caffeine in sport . influence of endurance exercise on the urinary caffeine concentration . 

 ✦ supervised fitness walking in patients with osteoarthritis of the knee . a randomized , controlled trial . 

 ✱ the effect of training intensity on ratings of perceived exertion . |

## DATASET

Obtaining labeled training data for conversation summarization is challenging. The available ones are either extractive (Verberne et al. (2018)) or too small (Barker et al. (2016); Anguera et al. (2012)) to train a neural model. To get around this issue and thoroughly verify the proposed architecture, we synthesized a dataset by utilizing a corpus of conventional texts for which summaries are available. We create two corpora of interleaved texts: one from the abstracts and titles of articles from the PubMed corpusand one from the questions and titles of Stack Exchange questions. A random interleaving of sentences from a few PubMed abstracts or Stack Exchange questions roughly resembles interleaved texts, and correspondingly interleaving of titles resembles its multi-sentence summary.

The algorithm that we devised for creating synthetic interleaved texts is defined in detail in the Appendix. The number of abstracts to include in the interleaved texts is given as a range (from $a$ to $b$) and the number of sentences per abstract to include is given as a second range (from $m$ to $n$). We vary the parameters as below and create three different corpora for experiments: **Easy** ($a$=2, $b$=2, $m$=5 and $n$=5), **Medium** ($a$=2, $b$=3, $m$=2 and $n$=5) and **Hard** ($a$=2, $b$=5, $m$=2 and $n$=5). Table 2 shows an example of a data instance in the Hard Interleaved RCT corpus.

## EXPERIMENTS

we report ROUGE-1, ROUGE-2, and ROUGE-L as the quantitative evaluation of the models. The Pubmed Medium and Hard corpora train, test and validation have approximately 290k, 6k and 1.5k instances, respectively. The Stack-Exchange Medium corpus train has 180k and Hard has 210k instances, while the both corpora have test and validation of approximately 5k and 4k instances, respectively. The remaining hyper-parameters are described in detail in the Appendix.

## BASELINES

We first trained and tested it on the Easy corpus. We also ran Shang et al. (2018)'s unsupervised two-step system on the test set of the Easy Interleaved Pubmed corpus. Additionally, we also utilized Shang et al. (2018)'s clustering component to first cluster the interleaved texts of the corpus, and then the disentangled corpus is used to train the seq2seq model. We refer to the latter as cluster→seq2seq. The performance comparison of Shang et al. (2018) and the two seq2seq models are shown in Table 3. Clearly, seq2seq performs better than Shang et al. (2018), the reason being a seq2seq model trained on a sufficiently large dataset is better at summarization than the unsupervised sentence com-

Table 3: Rouge F1-Scores for seq2seq models on the Pubmed Easy Corpus.

| Model | Rouge-1 | Rouge-2 | Rouge-L |
|---|---|---|---|
| Shang et al. (2018) | 30.37 | 10.77 | 20.04 |
| seq2seq | **44.38** | **19.47** | **35.20** |
| cluster→seq2seq | 42.93 | 18.76 | 30.68 |

Table 4: Recalls on a sampled Pubmed Hard-Corpus. Int=Interleaved, Dis=Disentangled.

| Model | type | Rouge-1 | Rouge-2 | Rouge-L |
|---|---|---|---|---|
| seq2seq | Int | 32.40 | 11.09 | 22.97 |
| hier2hier | Int | 33.51 | 12.50 | 24.27 |
| seq2seq | Dis | 37.01 | 14.44 | 26.91 |
| hier2hier | Dis | **37.68** | **15.11** | **28.25** |

pression (extractive) method. The lower performance of cluster→seq2seq in comparison to seq2seq shows that a disentanglement component is unnecessary in easy scenario.Furthermore, we utilize the ground-truth labels and disentangle the interleaved posts of a sampled (150k) Hard Corpus, and re-run the seq2seq model. The results (see Table 4) show seq2seq (row 3) takes advantage of cluster labels but hier2hier (row 4) does more than it.

## SEQ2SEQ VS. HIER2HIER MODELS

We then compare the proposed hierarchical approach against the seq-to-seq approach in summarizing the interleaved texts by experimenting on the Medium and Hard corpora.

Table 5 shows the experimental results on several corpora; clearly, an increase in the complexity of interleaving scantly impacts the performances. Though, in case of Pubmed, seq2seq depreciates by 1-2 points while hier2hier remains nearly same with increase of complexity. In Stack Exchange corpora, as Hard corpora has more training data (+30k) seq2seq improves slightly. However, a noticeable improvement is observed on changing the decoder to hierarchical, i.e., 1.5-3 Rouge points in Pubmed and 2-4.5 points in Stack Exchange depending on the interleaving complexity.

Table 5: Rouge Recall-Scores of models on the Medium and Hard Corpus.

| Corpus Difficulty | Model | Pubmed | | | Stack Exchange | | |
|---|---|---|---|---|---|---|---|
| | | Rouge-1 | Rouge-2 | Rouge-L | Rouge-1 | Rouge-2 | Rouge-L |
| Medium | seq2seq | 30.67 | 11.71 | 23.80 | 18.78 | 03.52 | 14.73 |
| | hier2hier | **32.78** | **12.36** | **25.33** | **24.34** | **05.07** | **18.63** |
| Hard | seq2seq | 29.07 | 10.96 | 21.76 | 20.21 | 04.03 | 14.93 |
| | hier2hier | **33.36** | **12.69** | **24.72** | **24.96** | **05.56** | **17.95** |

To understand the impact of hierarchy on the hier2hier model, we perform an ablation study and use the Hard Pubmed corpus for the experiments, and Table 6 shows the results. Clearly, adding hierarchical decoding already provides a boost in the performance. Hierarchical encoding also adds some improvements to the performance; however, the enhancement attained in training and inference speed by the hierarchical encoding is much more valuable. We will discuss it in depth later.

Further, we also compiled corpora to mimic real world conversation interleaving, wherein the sequence of summaries for interleaved posts may not follow the sequential occurrence of posts, see 17-18 in Algorithm. 1). An example, a post mention an action in the very beginning of conversation, but does only elaborates it at the end. Therefore, summary corresponding to the action should be at the end. So, we again create Medium and Hard real world mimicking corpora of the Stack Exchange dataset as it is more similar to real world conversations, and perform the same set of experiments. As seen in Table 7, the results show that both the seq2seq and hier2hier models performance are slightly lower as it is tougher task; see Table 5. In addition, the hier2hier model is still consistently better than the seq2seq model.

Importantly, hier2hier converges much earlier than the seq2seq and also reaches a lower training loss (Figure 5 in Appendix). hier2hier takes ≈1.5 days for training on a Tesla V100 GPU, while seq2seq takes ≈4.5 days. Therefore, the hier2hier model not only achieves greater accuracy but also reduces training and inference time.

Table 6: Rouge Recall-Scores of models on the Pubmed Hard Corpus.

| Model | Pubmed Hard Corpus | | |
|---|---|---|---|
| | Rouge-1 | Rouge-2 | Rouge-L |
| seq2seq | 29.07 | 10.96 | 21.76 |
| seq2hier | 32.92 | 11.87 | 24.43 |
| hier2seq | 31.86 | 11.9 | 23.57 |
| hier2hier | **33.36** | **12.69** | **24.72** |

Table 7: Rouge Recall-Scores of models on the Stack Exchange Medium and Hard Corpus.

| Model | Medium Corpus | | |
|---|---|---|---|
| | Rouge-1 | Rouge-2 | Rouge-L |
| seq2seq | 19.67 | 03.88 | 15.37 |
| hier2hier | **23.97** | **05.63** | **18.75** |
| | Hard Corpus | | |
| seq2seq | 19.62 | 03.71 | 14.90 |
| hier2hier | **24.14** | **05.00** | **17.25** |

## HIERARCHICAL ATTENTION

The contribution of the post-level and high-level attentions in a hierarchical decoder is two-fold: computing the thread representation and rescaling the word-level attentions. To understand the impact of hierarchical attention on the hier2hier model, we perform an ablation study of post-level attentions ($\gamma$) and high-level attentions ($\beta$), using the Hard corpus for the experiments.

Table 8: Rouge Recall-Scores of ablated models on the Hard Pubmed Corpus.

| Model | Rouge-1 | Rouge-2 | Rouge-L |
|---|---|---|---|
| hier2hier($+\gamma + \beta$) | **33.36** | **12.69** | **24.72** |
| hier2hier($-\gamma + \beta$) | 32.65 | 12.21 | 24.23 |
| hier2hier($+\gamma - \beta$) | 31.28 | 10.20 | 23.49 |
| hier2hier(Li,Luong&Jurafsky) | 29.83 | 09.80 | 22.17 |
| hier2hier($-\gamma - \beta$) | 30.58 | 10.00 | 22.96 |
| seq2seq | 29.07 | 10.96 | 21.76 |

Table 8 shows the performance comparison. Clearly, $\gamma$ attention improves the performance (0.5-1) of hierarchical decoding but not a lot. The high-level attention, i.e., $\beta$ is very important as without it the model performances is noticeably reduced (Rouge values decrease from 2-3). Additionally, we also include Li, Luong and Jurafsky, 2015 post-level attention technique in the comparison, as it is the closest hierarchical attention to ours (see Table 1). Though, (Li, Luong and Jurafsky) utilize a softmax obtained $\beta$ to compute thread representation, and in contrast to us, they do not reuse $\beta$ for re-scaling $\alpha$, and thereby, closest to hier2hier($+\gamma - \beta$). Evidently, re-utilization gives hier2hier($+\gamma - \beta$) an edge over (Li, Luong and Jurafsky, 2015). Lastly, removing both the $\gamma$ and $\beta$) makes the hier2hier similar to seq2seq, except a few more parameters, i.e., two additional LSTM, and the performance is also very similar.

## DISCUSSION

Table 9 shows an output of our hierarchical abstractive system, in which interleaved texts are in the top, and ground-truth and generated summaries in the bottom. Table 9 also shows the top two post indexes attended to by the post-level attention ($\gamma$) while generating those summaries, and they coincide with relevant posts. Similarly, the top 10 indexes (words) of the high-level attention ($\beta$)

Table 9: Interleaved sentences of 3 articles, and corresponding ground-truth and hier2hier generated summaries. The top 2 sentences that were attended ($\gamma$) for the generation are on the left. Additionally, top words ($\beta$) attended for the generation are colored accordingly.

| | Interleaved Texts |
|---|---|
| 0 | this study was conducted **to evaluate the influence of** excessive **sweating** during **long-distance** running **on** the urinary concentration of **caffeine**... |
| 1 | **to assess the effect of** a **program of** supervised **fitness** walking and patient education on functional status , pain , and... |
| ... | ... |
| 5 | a total of 102 patients with a documented diagnosis of primary osteoarthritis of one or both knees participated... |
| 6 | we **examined the effects of intensity of training** on ratings of perceived exertion (... |
| ... | ... |
| | GroundTruth/Generation |
| 0,2 | caffeine in sport . influence of endurance exercise on the urinary caffeine concentration . **effect of excessive [UNK] during [UNK] running on the urinary concentration of caffeine .** |
| 1,4 | supervised fitness walking in patients with osteoarthritis of the knee . a randomized , controlled trial . **effect of a physical fitness walking on functional status , pain , and pain** |
| 6,8 | the effect of training intensity on ratings of perceived exertion . **effects of intensity of training on perceived [UNK] in [UNK] athletes .** |

is directly visualized in the table through the color coding matching the generation. The system not only manages to disentangle the interleaved texts but also to generate appropriate abstractive summaries. Meanwhile, $\beta$ provides explainability of the output.

As interleaving in the table includes abstracts from the same category, e.g. Physical Activities, the interleaving is complex and approximates the real-world conversations. Despite that, the end-to-end hierarchical system tackles the task to a large extent. The next, future stage in this research is transfer learning of the hierarchical system to more types of conversations. We find models trained for 50k iteration on Pubmed Hard corpora (a=6, b=10, m=2 and n=3) and then fine-tuned on the popular meeting AMI corpus (McCowan et al. (2005)) already reaches competitive results (Rouge-1=39.81, Rouge-2=11.35) against SOA system on AMI (Shang et al. (2018) Rouge-1=37.86 and Rouge-2=07.84). Additionally, we aim to modify hier2hier to include some of the recent additions of seq2seq models, e.g., See et al. (2017) pointer mechanism.

## CONCLUSION

We presented an end-to-end trainable hierarchical encoder-decoder architecture which implicitly disentangles interleaved texts and generates a multi-sentence abstractive summary covering the text threads. Furthermore, the architecture addresses the error propagation issue that occurs in the two-step architectures. Our proposed novel hierarchical attention further boosts both disentanglement and summary generation.

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

# APPENDIX

## INTERLEAVE ALGORITHM

In Algorithm. 1, INTERLEAVE takes a set of concatenated abstracts and titles, $C = \langle A_1; T_1, \ldots, A_{|C|}; T_{|C|} \rangle$, minimum, $a$, and maximum, $b$, number of abstracts to interleave, and minimum, $m$, and maximum, $n$, number of sentences in a source, and then returns a set of concatenated interleaved texts and summaries. WINDOW takes a sequence of texts, $X$, and returns a window iterator of size $\frac{|X|-w}{t} + 1$, where $w$ and $t$ are window size and sliding step respectively. *window* reuses elements of $X$, and therefore, enlarges the corpus size. Notations $\mathcal{U}$ refers to a uniform sampling, $[\cdot]$ to array indexing, and REVERSE to reversing an array.

---

**Algorithm 1** Interleaving Algorithm

---

1: **procedure** INTERLEAVE($C, a, b, m, n$)
2: $\quad \hat{C}, Z \leftarrow$ WINDOW($C, w = b, t = 1$), Array()
3: $\quad$ **while** $\hat{C} \neq \emptyset$ **do**
4: $\quad\quad C', A', T', S \leftarrow \hat{C}$.NEXT(), Array(), Array(), {}
5: $\quad\quad r \sim \mathcal{U}(a, b)$
6: $\quad\quad$ **for** $j = 1$ to $r$ **do** $\qquad\qquad\qquad\qquad\qquad\qquad\qquad\triangleright$ Selection
7: $\quad\quad\quad A, T \leftarrow \hat{C}[j]$
8: $\quad\quad\quad T'$.ADD($T$)
9: $\quad\quad\quad q \sim \mathcal{U}(m, n)$
10: $\quad\quad\quad A'$.ADD($A[1{:}q]$)
11: $\quad\quad\quad S \leftarrow S \cup \{j_{\times q}\}$
12: $\quad\quad \hat{A}, \hat{T} \leftarrow$ Array(), Array()
13: $\quad\quad$ **for** 1 to $|S|$ **do** $\qquad\qquad\qquad\qquad\qquad\qquad\triangleright$ Interleaving
14: $\quad\quad\quad k \leftarrow \mathcal{U}(S)$
15: $\quad\quad\quad S \leftarrow S \backslash k$
16: $\quad\quad\quad I \leftarrow$ REVERSE($A'[k]$).POP()
17: $\quad\quad\quad \hat{A}$.ADD($I$)
18: $\quad\quad\quad J \leftarrow T'[k]$
19: $\quad\quad\quad$ **if** $J \notin \hat{T}$ **then**:
20: $\quad\quad\quad\quad \hat{T}$.ADD($J$)
21: $\quad\quad Z$.ADD($\hat{A}; \hat{T}$)
22: $\quad$ **return** $Z$

---

## PARAMETERS

For the word-to-word encoder, the steps are limited to 20, while the steps in the word-to-word decoder are limited to 14. The steps in the post-to-post encoder and thread-to-thread decoder depend on the corpus type, e.g., Medium has 15 steps in post-to-post and 3 steps in thread-to-thread. In seq2seq experiments, the source is flattened, and therefore, the number of steps in the source is limited to 300. We initialized all weights, including word embeddings, with a random normal distribution with mean 0 and standard deviation 0.1. The embedding vectors are of dimension 80. The vocabulary size is limited to 5000 and 15000 for Pubmed and Stack Exchange corpora respectively. All hidden states of the encoder and decoder in the models are set to dimension 100. We pad short sequences with a special token, $\langle PAD \rangle$. We use Adam Kingma & Ba (2014) with an initial learning rate of .0001 and batch size of 64 for training. Texts are lowercased and numbers are replaced by the special symbol $\%$.

## DATA POINTS AND RESULTS

## TRAINING: SEQ2SEQ VS HIER2HIER

Table 10: The left rows contain interleaving of 3 articles with 2 to 5 sentences and the right rows contain their interleaved titles. Associated sentences and titles are depicted by similar symbols.

| | |
|---|---|
| ✓ botulinum toxin a is effective for treatment. . . 
 ✓ the trigone is generally spared because of the theoretical. . . 
 ✓ evaluate efficacy and safety of trigone-including . . . . 
 ✦ most methadone-maintained injection drug users . . . 
 ✱ gender-related differences in the incidence of bleeding. . . 
 ✱ we studied patients with stemi receiving fibrinolysis. . . 
 ✦ physicians may be reluctant to treat hcv in idus because . . . 
 ✱ outcomes included moderate or severe bleeding defined . . . 
 ✦ optimal hcv management approaches for idus remain . . . 
 ✱ moderate or severe bleeding was 1.9-fold higher . . . 
 ✦ we are conducting a randomized controlled trial in a network. . . 
 ✱ bleeding remained higher in women even after adjustment 
 . . . | ✓ prospective randomised controlled trial comparing trigone-sparing versus trigone-including intradetrusor injection of abobotulinumtoxina for refractory idiopathic detrusor overactivity. 

 ✦ rationale and design of a randomized controlled trial of directly observed hepatitis c treatment delivered in methadone clinics. 

 ✱ comparison of incidence of bleeding and mortality of men versus women with st-elevation myocardial infarction treated with fibrinolysis. |

Table 11: Interleaved sentences of 3 articles, and corresponding ground-truth and hier2hier generated summaries. The top 2 sentences that were attended ($\gamma$) for the generation are on the left. Additionally, top words ($\beta$) attended for the generation are colored accordingly.

| | Interleaved Texts |
|---|---|
| 0 | **botulinum toxin** a is **effective** for **treatment** of **idiopathic detrusor** overactivity ( [UNK] ) |
| 1 | the [UNK] is generally [UNK] because of the theoretical risk of [UNK] reflux ( [UNK] ) , although studies assessing. . . |
| . . . | . . . |
| 3 | **most [UNK] injection drug users** ( **idus** ) have been infected with hepatitis c virus ( hcv ) , but. . . |
| 4 | **[UNK] differences** in the **incidence** of **bleeding** and its relation to subsequent mortality in patients with st-segment elevation myocardial infarction. . . |
| . . . | . . . |
| 8 | **optimal** hcv management approaches for idus remain unknown . . . . |
| . . . | . . . |
| | GroundTruth/Generation |
| 0,1 | prospective randomised controlled trial comparing trigone-sparing versus trigone-including intradetrusor injection of abobotulinumtoxina for refractory idiopathic detrusor overactivity. 
 **efficacy of [UNK] [UNK] in patients with idiopathic detrusor overactivity : rationale , design** |
| 3,4 | rationale and design of a randomized controlled trial of directly observed hepatitis c treatment delivered in methadone clinics. 
 **validation of a point-of-care hepatitis injection drug injection drug , hcv medication , and** |
| 4,8 | comparison of incidence of bleeding and mortality of men versus women with st-elevation myocardial infarction treated with fibrinolysis . 
 **subgroup analysis of patients with st-elevation myocardial infarction with st-elevation myocardial infarction .** |

Table 12: The left rows contain interleaving of 4 articles with 2 to 5 sentences and the right rows contain their interleaved titles. Associated sentences and titles are depicted by similar symbols.

| | |
|---|---|
| ✓ the effects of short-course antiretrovirals given to...
✱ good adherence is essential for successful antiretroviral...
✓ women in kenya received short-course zidovudine ( zdv )...
✓ breast milk samples were collected two to three times weekly....
✦ the present primary analysis of antiretroviral therapy with...
✱ this was an observational analysis of an open multicenter...
✦ patients with hiv-1 rna at least 5000 copies/ml were...
✱ at 4-weekly clinic visits , art drugs were provided and ...
✦ the primary objective was to demonstrate non-inferiority...
✱ viral load response was assessed in a subset of patients...
♣ we explored the link between serum alpha-fetoprotein levels...
✱ drug possession ratio ( percentage of drugs taken between...
♣ a low alpha-fetoprotein level ( < 5.0 ng/ml ) was an...
✦ six hundred and eighty-nine patients were randomized...
✦ at 48 weeks , 84 % of drv/r and 78 % of lpv/r patients...
✓ hiv-1 dna was quantified by real-time pcr ....
♣ serum alpha-fetoprote in measurement should be integrated ... | ✓ hiv-1 persists in breast milk cells despite antiretroviral treatment to prevent mother-to-child transmission.

✱ patterns of individual and population-level adherence to antiretroviral therapy and risk factors for poor adherence in the first year of the dart trial in uganda and zimbabwe.

✦ efficacy and safety of once-daily darunavir/ritonavir versus lopinavir/ritonavir in treatment-naive hiv-1-infected patients at week 48.

♣ serum alpha-fetoprotein predicts virologic response to hepatitis c treatment in hiv coinfected patients. |

Table 13: Interleaved sentences of 4 articles, and corresponding ground-truth and hier2hier generated summaries. The top 2 sentences that were attended ($\gamma$) for the generation are on the left. Additionally, top words ($\beta$) attended for the generation are colored accordingly.

| | Interleaved Texts |
|---|---|
| 0 | the **effects** of **short-course antiretrovirals given** to **reduce mother-to-child transmission** ( [UNK] ) on temporal patterns of [UNK] hiv-1 rna |
| 1 | **good adherence is essential for successful antiretroviral therapy** ( **art** ) provision , but simple measures have rarely been validated... |
| 2 | women in kenya received short-course zidovudine ( zdv ) , single-dose nevirapine ( [UNK] ) , combination [UNK] or short-course... |
| 3 | breast milk samples were collected two to three times weekly for 4-6 weeks .... |
| 4 | the **present primary analysis** of **antiretroviral therapy with  [UNK] examined in** naive subjects ( [UNK] ) compares the efficacy and... |
| ... | ... |
| 10 | we **explored the link between serum [UNK]** levels and virologic response in [UNK] [UNK] c virus coinfected patients .... |
| ... | ... |

| | GroundTruth/Generation |
|---|---|
| 0,2 | hiv-1 persists in breast milk cells despite antiretroviral treatment to prevent mother-to-child transmission .
**impact of hiv-1 persists on hiv-1 rna in human immunodeficiency virus-infected individuals with hiv-1** |
| 1,3 | patterns of individual and population-level adherence to antiretroviral therapy and risk factors for poor adherence in the first year of the dart trial in uganda and zimbabwe .
**impact of a antiretroviral treatment algorithm on adherence to antiretroviral therapy in [UNK] ,** |
| 4,2 | efficacy and safety of once-daily darunavir/ritonavir versus lopinavir/ritonavir in treatment-naive hiv-1-infected patients at week 48 .
**a randomized trial of [UNK] versus [UNK] in treatment-naive hiv-1-infected patients with hiv-1 infection** |
| 10,12 | serum alpha-fetoprotein predicts virologic response to hepatitis c treatment in hiv coinfected patients .
**predicting virologic response in [UNK] coinfected patients coinfected with hiv-1 : a [UNK] randomized** |

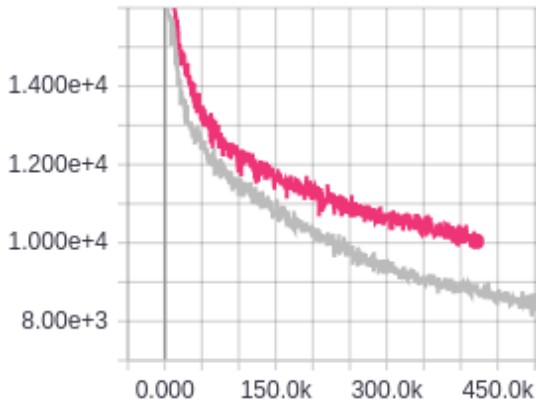

Figure 5: A comparison of running average training loss between seq2seq (pink) and hier2hier (gray) for Stack Exchange Hard corpus.

