# OpenReview forum: "Generating Multi-Sentence Abstractive Summaries of Interleaved Texts"
_ICLR.cc/2020/Conference — Reject_

### Official Review · AnonReviewer1 · 2019-10-21
**Official Blind Review #1786**

**Rating:** 3

**Review:**

<Strengths>
+ This paper addresses the problem of summarizing interleaved posts, which involves two subtasks of identifying the clusters of threads and summarizing each thread into a sentence.
+ This is an understudied area of research in summarization research, only few previous works have been done; for example, (Shang et al. 2018). The proposed approach is end-to-end  trainable, whereas (Shang et al. 2018)’s method tackles the problem in two separate stages.

<Weakness>
1. The technical novelty may need to be further justified.
- The proposed model (in Fig.2) consists of the encoder, the decode and hierarchical attention module.
- The encoder and the decoders are standard ones based on (Nallapati et al. 2017) and (Li et al. 2015), respectively. It is hard to find any technical novelty here.
- This paper argues that the proposed hierarchical attention is novel with no clear ground. It simply mentions the inspiration from (See et al. 2017), but does not explicitly what aspects are novel as Hierarchical attention has been popularly used in general seq2seq models for many NLP applications.
- Table 1 is rather cursory and incomplete as it compares with only three different models.

2. Experiments have much room for improvement.
- The two datasets are simply mixture of text from existing datasets and rather artificial and far from actual interleaved posts, as shown in Table 2.
- In page 6, three different corpora are introduced with varying interleave parameters. However, the parameters a,b,m, and n are rather small (<= 5), so it is hard to say the configurations are challenging.
- The SOTA model (Shang et al. 2018) is only compared on the PubMed Easy Corpus in Table 3. Is there any special reason to exclude it from the comparison with more challenging corpora?
- In most results of experiments, the paper focus on showing that the proposed hierarchical attention model is better than basic seq2seq, which may not be strong results to show the effectiveness of the proposed model.

<Minor comments>
- This draft does not seem to follow the ICLR format.
- Fig.2-4 are hard to see in details. They should be enlarged and re-organized.
- References should be added to Table 1.
- This paper exceeds the limit of 8 pages. I cannot find any good strong ground to have more additional pages than the commended length.

<Conclusion>
My rate is borderline with slightly leaning toward reject, mainly due to issues of technical novelty and evaluation. I will decide my rate finally, based on the authors’ response to the concerns above.


**Experience Assessment:**

I have published one or two papers in this area.

**Review Assessment: Checking Correctness Of Derivations And Theory:**

I assessed the sensibility of the derivations and theory.

**Review Assessment: Checking Correctness Of Experiments:**

I carefully checked the experiments.

**Review Assessment: Thoroughness In Paper Reading:**

I read the paper thoroughly.

---

> ### Author Response · Authors · 2019-11-12
> **performance on synthetic and real world dataset**
>
> Thank you for your careful review. Regarding point 1 in the weakness, we would like to inform the reviewer that we have updated the hierarchical attention section, a key contribution, so, it is more clear and differentiable from other existing methodologies. To summarize, as ours is the first attempt in applying hierarchical decoding with hierarchical attention in the domain of text-summarization, we found that the previous hierarchical attention approach by Nallapati et al. 2016, i.e., using sentence (post) level attention (gamma) to scale word level attention (alpha) didn't improve very much, i.e., Rouge-1 of 31.3 and Rouge-2 of 10.2 against Rouge-1 of 30.6 and Rouge-2 10.0,  in the hierarchical decoding of interleave text-summaries; see scores in rows 3 and 5 in Table 8. Therefore, in contrast to earlier work, we introduce another level of attention, i.e., high-level attentions (beta). Our newly introduced beta attention is also computed at the word-level with the primary objective being to obtain a thread representation and secondarily to rescale word-attention (alpha). Additionally, beta attention is rescaled by post attention (gamma). Thus, the major difference in our novel approach is attention rescaling, i.e., gamma->beta->alpha, where ‘->’ indicates rescaling,  in contrast to previous approaches (Nallapati et al, Cheng et al. 2016, Tan et al. 2017 and Hsu et al. 2018), which use gamma->alpha only. The scores in rows 1 (Rouge-1 33.4) and 2 (Rouge-1 32.7) in Table 8 show it improves performance against previous approach (Rouge-1 31.3), and examples in Table 9 and the Appendix show beta attention brings in phrase-level information and also works in tandem with gamma.
>
> Regarding novelty in encoder and decoder, we would like to inform the reviewer that Nallapati et al. 2017 used a hierarchical encoder, they did not use a hierarchical decoder. We are the first to attempt hierarchical decoding in text-summarization while Li et al. 2015 applied it in the context of Autoencoders. Hierarchical decoding not only improves performance, but also computation speed which is equally important in case of longer summaries.
> We believe summarization is more challenging since it requires generating a new text distilling the original; it also forms a real test bed for hierarchical decoding.
>
> Regarding novelty in hierarchical attention, Thanks for letting us know the lacking, we have updated Table 1 with citations and also updated the hierarchical attention section. We refer the reviewer to page 6. Further, we draw inspiration from the See et al. 2017 pointer mechanism which allows modelling two distributions, i.e., over source tokens and vocabulary tokens together using a switch probability. Basically, the intuition is that in an end-to-end learning if there are two distributions that involves a common set of tokens then they must be modeled together by something, e.g., switch probabilities, but not in isolation. We also model two distributions, but in contrast to See et al., we use them to capture larger-scale and local context, i.e., high-level distribution (beta) and low-level distribution (alpha) over the same source tokens and we model them together by scaling one (alpha) by the other (beta). Further, the popularly held intuition of hierarchical attention, i.e., sentence attention scales word attention, is still upheld as gamma (post-attention) scales the beta. Therefore, all attentions are modeled together without any extra train labels unlike Hsu et al 2018.
>
> Regarding point 2 in the weakness, the reason for exclusion is that the disentanglement component of (Shang et al. 2018) requires an explicit number of clusters, which fixes the number of summaries, while the medium and hard corpus has a variable number of clusters, i.e., 2 to 3 (medium) and 2 to 5 (hard). We believe our hard corpus is a challenging task as the model has to detect highly variable threads, i.e., 2 to 5 from 500 source tokens. To justify this, we would also like to note that we mention in the Discussion section that we ran our model on the AMI corpus, a real-world meeting corpus, where the source size was limited to 600 and target to 150 tokens and hier2hier gives us Rouge-1 of 37.57 and Rouge-2 of 11.72, which outperforms the results of Shang et al. 2018 (approx Rouge-1 of 30) for the same target size (150 tokens). Additionally, we fine-tuned one of the Pubmed trained models to the AMI corpus and found a simple fine-tuning after 50k iterations lead to improvement of an additional 2-3 rouge scores.
> As these experiments were not part of the original submission, we can update the paper with them upon the acceptance. Regarding the last point in the weakness, we would like to inform reviewer that it is not only the novel hierarchical attention but also hierarchical decoding that is adding gain in the performance.
>
> Further, we have reformatted images and tables, and reduced the draft size to a little over 8 page.

---

### Official Review · AnonReviewer2 · 2019-10-22
**Official Blind Review #2**

**Rating:** 3

**Review:**

This paper proposes a technique for generating summaries of interleaved texts. Unlike previous work that first perform unsupervised clustering to extract ordered threads, the authors instead propose a hierarchical model that directly process interleaved threads. In particular, the authors propose using a hierarchical encoder that encodes words -> post -> threads as well as a hierarchical decoder that decodes threads -> words. On synthetic interleaved datasets composed from PubMed and StackExchange, the proposed method outperform seq2seq with attention as well as a pipeline system that applies unsupervised clustering followed by seq2seq. The authors present results for well-chosen ablations as well as baselines.

My concerns stem from the synthetic nature of these tasks. In particular:

- is it difficult to disentangle the threads? Suppose we train a simple, supervised model instead of using unsupervised clustering and run the same pipeline experiment, how well would the model perform compared to hier2hier?
- due to the nature of these two tasks, is disentanglement even helpful for summarization? If not, heir2hier might be picking up on helpful signals that are not accessible to baselines. I don't have a good idea of what the tasks look like from reading the paper. I would ask that the authors put some examples of input output pairs in the paper. The algorithm in the appendix for interleaving is not very helpful.
- I would like to see some upperbound of running seq2seq on ground-truth disentangled threads.

Some comments on the writing:
- Figure 4 and Table 2 are way too small

**Experience Assessment:**

I have published in this field for several years.

**Review Assessment: Checking Correctness Of Derivations And Theory:**

I assessed the sensibility of the derivations and theory.

**Review Assessment: Checking Correctness Of Experiments:**

I assessed the sensibility of the experiments.

**Review Assessment: Thoroughness In Paper Reading:**

I read the paper at least twice and used my best judgement in assessing the paper.

---

> ### Author Response · Authors · 2019-11-12
> **upper bound in interleaved text summarization**
>
> Thanks for the valuable feedback. To address some of the reviewer concerns and questions, we ran the experiments to examine performance when disentangled threads are concatenated and used as input, e.g., posts-thread1->posts-thread2->posts-thread3->posts-thread4 => summary1->summary2->summary3->summar4, and for expediency, we used a 150k sampled Pubmed Hard corpus as the original Pubmed Hard corpus is twice the size and a seq2seq model takes about 5 days to converge. The results are presented in Table 4. Evidently, seq2seq doesn't gain much from ground-truth disentangled posts, which is coherent with the result for disentangling using clustering on the easy corpus (see Table 3). We believe this is due to the sequential decoder not being able to utilize the later cluster of posts. Consider a case where all the sentences in a channel are very similar, e.g., all discussing HIV (see example 2 in the Appendix), and when the seq2seq encoder runs over such disentangled posts sequentially, i.e., cluster1-cluster2-cluster3-cluster4 and the decoder generates summaries sequentially, i.e., summary1-summary2-summary3-summary4, then it might be good at generating  summary1 and summary2; however, it may not be as good for  summary3 and summary4, due to the long-term dependency problem.
>
> Additionally, Table 4 shows that hier2hier still performs best when the posts are disentangled. Also, comparison of hier2hier results on interleaved and disentangled corpora shows that hier2hier is quite successful in disentangling on its own. see examples in the Table 9 and appendix. Overall, disentanglement is helpful for hier2hier models, however it’s a time-consuming process and can lead to error propagation.
>
> Regarding concerns on how tasks look like, we have rearranged Table 2 which illustrates an interleaved text in the corpus, and  Table 9, which illustrates model output. We hope this makes the task further understandable . Additionally, we have put more examples in the appendix. Further, we also modified the draft to address reviewer concern on images and tables formatting, they are either enlarged, rearranged or combined to enhance comprehensibility.

---

> > ### Author Response · Authors · 2019-11-14
> > **updated experiments**
> >
> > We would like to let the reviewer know that there was a mix up while running the disentangled text experiments:  an incorrectly sampled corpus was used. Therefore, we reran the disentangled experiments with the correct 150k samples of the hard Pubmed corpus and updated the results (see Table 4). The results indicate that a sequential decoder is utilizing the ground-truth cluster information (concatenated disentangled threads), and therefore, is an upper-bound. However, hier2hier is better than seq2seq in utilizing ground-truth cluster information (see row 3 and 4 of Table 4).

---

### Official Review · AnonReviewer3 · 2019-10-28
**Official Blind Review #3**

**Rating:** 6

**Review:**

This work proposes an architecture to generate summaries for multi-participant postings as in chat conversations. Previous work tackled this problem as a pipeline of disentangling threads from mixed chat posts and then generating a summary for each post. The presented work proposes a hierarchical attention model to solve this task in an end-to-end fashion.

The proposed architecture is divided into :

- a hierarchical encoder: encodes each sentence to a vector using bi-lstm and average pooling which is then fed to a post-to-post encoder which generates a set of representations corresponding to those posts.

- a hierarchical decoder: number of disentangled posts are generated using a thread-to-thread decoder which given the max pooled representations from the post-to-post encoder generated a number of initial representations for each thread summary using hierarchical attention combining attention over paragraphs and over words at each decoding time step.

Experiments:
Authors demonstrate the effectiveness of their approach by comparing against a set of baselines traditional seq2seq, as well as cluster-> seq2seq and the model from Shang et al. (2018). Evaluation datasets consisted of several datasets for chat summarization synthesized to mimic interleaved conversation with different difficulties.

Through an ablation study, authors also show the effectiveness of the hierarchical encoding and hierarchical decoding as well as the post level and high-level attention.

overall:
Although this paper is well written and well-motivated. The work is a bit incremental and builds upon previous ideas from hierarchical attention literature to apply for interleaved text generation. The provided baselines are quite weak compared to the SOTA summarization methods at the moment, although none of them is directly modelled for interleaved text summarization through multi summaries.

typos:
* Nallapati et al. Nallapati et al. (2016)
* See et al. See et al. (2017)


Question:
Is there an intuition behind penalizing a model that generates correct summaries for each thread with different order? Do references summaries follow the same order as in the threads in the conversation?

In the paper you dedicated several paragraphs to reimplement basic models such as seq2seq or (See et al.) pointer generator network and make sure they work correctly. There are existing implementations and pre-trained models available such as in OpenNMT.

**Experience Assessment:**

I have published one or two papers in this area.

**Review Assessment: Checking Correctness Of Derivations And Theory:**

I carefully checked the derivations and theory.

**Review Assessment: Checking Correctness Of Experiments:**

I carefully checked the experiments.

**Review Assessment: Thoroughness In Paper Reading:**

I read the paper at least twice and used my best judgement in assessing the paper.

---

> ### Author Response · Authors · 2019-11-12
> **elaborated the novelty in hierarchical attention**
>
> Thanks for the valuable feedback.
> Regarding the reviewers first concern on novelty in hierarchical attention, we agree that there are several pieces of literature on hierarchical attention for NLP tasks. Since we are focused on text-summarization, Table 1 (updated) summarizes some of the forms of hierarchical attention in the domain. We would also like to point out that previous ideas on hierarchical attention in the field were devised assuming sequential decoding, and top-down integration of attention, where sentence- (post-) level attention scale word-level attentions, e.g., Nallapati et al. and Hsu et al. As far as we know, ours is the first architecture of applying hierarchical decoding with hierarchical attention in text-summarization, and we found the previous approach on hierarchical attention, i.e., sentences level attentions scaling word level doesn't help much in hierarchical decoding of interleaved text; see scores in row 3 (hier2hier(+gamma-beta)) and row 5 (hier2hier(-gamma-beta)) in Table 7. Therefore, we introduce another mid-level attention, and refer to it as high-level attention (beta), which brings in phrase-level information and works in tandem with both post- and word-level attentions. We refer the reviewer to the updated hierarchical attention section on page 5.
>
> Regarding the second concern on weak baseline, we agree that SOTA summarization is superior to the base model, i.e., seq2seq; however, most of these SOTA are also built on seq2seq with advance features like a pointer mechanism and/or coverage and/or RL training. As future work, we aim to include those in our hier2hier setup, as it improved the performance in the news summarization task, we believe the similar improvement will follow in the interleaved case.
>
> Further, we find a very good point is raised in the reviewer first question. At the moment, due to way seq2seq is trained, we are forcing our model to learn to generate tokens by recognizing the occurrence of thread (Table 5) or
> the density of threads (Table 7) as this is how token of summaries arranged; however, in the future, we will design or use a training loss that is computed at the summaries level rather than tokens. Further, we would also like to point out that the ROUGE-1 scores; however, doesn’t penalize the order of summaries, and the number of transitions between summaries is much smaller than the number of transitions between word pairs within a summary.
> Regarding the second question, the answer is yes. We have compiled the corpus in a manner that the reference summary order follows the first new thread occurrence, e.g.,
> post1-thread1, post1-thread2, post-2-thread2, post1-thread3 => thread1-summary, thread2-summary, thread3-summary. Additionally, we evaluated models strength in recognizing threads where summaries are ordered by the location of each thread’s greatest density. Here, density refers to smallest window of posts with over 50% of posts belonging to a thread; e.g., post1-thread1, post1-thread2, post-2-thread2, post2-thread1, post3-thread1, post4-thread1 => thread2-summary, thread1-summary. In this example, although thread1 occurs early, as the majority of posts on thread1 occurs latter, therefore, its summary also occurs later. As shown in Table 7, the model recognizes threads even in these scenarios, which is indicated by its performance (see Table 7) and is similar to the above scenario (Table 5).
> Regarding the concern on the implementation, we would like to inform reviewer that we indeed used one provided by TensorFlow, however, we modified hyperparameters and learning algorithms and therefore wanted to convey that the gains are not due to those changes. We will reference the TensorFlow model that we used as a base. Further, we have also modified the draft to fix typos pointed out by the reviewer and also changed images and tables formatting to enhance comprehensibility, now the draft size is a little over 8 page.

---

### Decision · Program_Chairs · 2019-12-19

**Decision:**

Reject

**Comment:**

This paper proposes an end-to-end approach for abstractive summarization of on-line discussions. The approach is contrary to the previous work that first disentangles discussions, and the summarizes them, and aims to tackle transfer of disentanglement errors in the pipeline. The proposed method is a hierarchical encoder - hierarchical decoder architecture. Experimental results on two corpora demonstrate the benefits of the proposed approach. The reviewers are concerned about the synthetic nature of the datasets, limited novelty given the previous work, lack of clear explanation of whether disentanglement is actually needed for summarization, and simpler baselines in comparison to the state-of-the-art. Hence, I recommend rejecting the paper.